# Pipe-SGD: A Decentralized Pipelined SGD Framework for Distributed Deep Net Training

**Youjie Li**[†], **Mingchao Yu**[*], **Songze Li**[*], **Salman Avestimehr**[*],
**Nam Sung Kim**[†], and **Alexander Schwing**[†]

[†]University of Illinois at Urbana-Champaign
[*]University of Southern California

## Abstract

Distributed training of deep nets is an important technique to address some of the present day computing challenges like memory consumption and computational demands. Classical distributed approaches, synchronous or asynchronous, are based on the parameter server architecture, *i.e.*, worker nodes compute gradients which are communicated to the parameter server while updated parameters are returned. Recently, distributed training with AllReduce operations gained popularity as well. While many of those operations seem appealing, little is reported about wall-clock training time improvements. In this paper, we carefully analyze the AllReduce based setup, propose timing models which include network latency, bandwidth, cluster size and compute time, and demonstrate that a pipelined training with a width of two combines the best of both synchronous and asynchronous training. Specifically, for a setup consisting of a four-node GPU cluster we show wall-clock time training improvements of up to $5.4\times$ compared to conventional approaches.

## 1 Introduction

Deep nets [25, 3] are omnipresent across fields from computer vision and natural language processing to computational biology and robotics. Across domains and tasks they have demonstrated impressive results by automatically extracting hierarchical abstractions of representations from many different datasets. The surge in popularity pivoted in the 2010s, with impressive results being demonstrated on the ImageNet dataset [22, 42]. Since then, deep nets have been applied to many more tasks. Prominent examples include recognition of places [53], playing of Atari games [34, 35], and the game of Go [45]. Common to all those methods is the use of large datasets to fuel the many layers of deep nets.

Importantly, in the last few years, the number of layers, or more generally the depth of the computation tree has increased significantly from a few layers for LeNet [26] to several 100s or 1000s [14, 24]. Inherent to the increasing complexity of the computation graph is an increase in training time and often also an increase in the amount of data that is processed. Traditionally, computational performance increases do not keep up with the desired processing needs despite the use of accelerators like GPUs.

Beyond accelerators, parallelization of computation on multiple computers is therefore popular. However, it requires frequent communication to exchange a large amount of data among compute nodes while the bandwidth of network interfaces is limited. This in turn significantly diminishes the benefit of parallelization, as a substantial fraction of training time is spent to communicate data. The fraction of time spent on communication is further increased when applying accelerators [16, 7, 38, 52, 48, 49, 44], as they decrease computation time while leaving communication time untouched.

To take advantage of parallelization across machines, a variety of approaches have been developed starting from the popular MapReduce paradigm [9, 51, 19, 37]. Despite their benefits, communication heavy training of deep nets is often based on custom implementations [8, 6, 36, 20] relying on the parameter server architecture [28, 27, 15], where the centralized server aggregates the gradients from workers and distributes the updated weights, either in a synchronous or asynchronous manner. Recent research proposed to use a decentralized architecture with global synchronization among nodes [12, 33]. However, in common to all the aforementioned techniques, little is reported regarding the timing analysis of distributed deep net training.

In this paper, we analyze the wall-clock time trade-offs between communication and computation. To this end we develop a model to assess the training time based on a set of parameters such as latency, cluster size, network bandwith, model size, *etc*. Based on the results of our model we develop Pipe-SGD, a framework with pipelined training and balanced communication, and show its convergence properties by adjusting proofs of [23, 15]. We also show what types of compression can be efficiently included in an AllReduce based framework. Finally, we assess the speedups of our proposed approach on a GPU cluster of four nodes with 10GbE network, showing wall-clock time training improvements by a factor of $3.2 \sim 5.4\times$ compared to conventional centralized and decentralized approaches without degradation in accuracy.

## 2 Background

**General Training of Deep Nets:** Training of deep nets involves finding the parameters $w$ of a predictor $F(x,w)$ given input data $x$. To this end we minimize a loss function $\ell(F(x,w),y)$ which compares the predictor output $F(x,w)$ for given data $x$ and the current $w$ to the ground-truth annotation $y$. Given a dataset $\mathscr{D} = \{(x,y)\}$, finding $w$ is formally summarized via:

$$\min_w f_{\mathscr{D}}(w) := \frac{1}{|\mathscr{D}|} \sum_{(x,y)\in\mathscr{D}} \ell(F(x,w),y). \tag{1}$$

Optimization of the objective given in Eq. (1) w.r.t. the parameters $w$, *e.g.*, via gradient descent using $\frac{\partial f_{\mathscr{D}}}{\partial w}$, can be challenging due to not only the complexity of evaluating the predictor $F(x,w)$ and its derivative, but also the size of the dataset $|\mathscr{D}|$. Consequently, stochastic gradient descent (SGD) emerged as a popular technique. We randomly sample a subset $\mathscr{B}$ of the dataset, often also referred to as a minibatch. Instead of computing the gradient on the entire dataset $\mathscr{D}$, we approximate it using the samples in the minibatch, *i.e.*, we assume $\frac{\partial f_{\mathscr{D}}}{\partial w} \approx \frac{\partial f_{\mathscr{B}}}{\partial w}$. However, for present day datasets and predictors, computation of the gradient $\frac{\partial f_{\mathscr{B}}}{\partial w}$ on a single machine is still challenging. Minibatch sizes $|\mathscr{B}|$ of less than 20 samples are common, *e.g.*, when training for semantic image segmentation [5].

**Distributed Training of Deep Nets:** To train larger models or to increase the minibatch size, distributed training on multiple compute nodes is used [8, 15, 6, 27, 28, 36, 16]. A popular architecture to facilitate distributed training is the parameter server framework [15, 27, 28]. The parameter server maintains a copy of the current parameters, and communicates with a group of worker nodes, each of which operates on a small minibatch to compute local gradients based on the retrieved parameters $w$. Upon having completed its task, the worker shares the gradients with the parameter server. Once the parameter server has obtained all or some of the gradients it updates the parameters using the negative gradient direction and afterwards shares the latest values with the workers.

Asynchronous updates where each worker independently pulls $w$ from the server, computes its own local gradient, and pushes results back are available and illustrated in Fig. 1 (a). Due to the asynchrony, minimal synchronization overhead is traded with staleness of gradients. Methods for staleness control exist, which bound the number of delay steps [15]. However, note that stale gradients may slow down training significantly.

Importantly, all those frameworks are based on a centralized compute topology which forms a communication bottleneck, increasing the training time as the cluster size scales. The time taken by pushing gradient, update, and pulling $w$ can be linear in the cluster size due to network congestion.

Therefore, most recently, decentralized training frameworks gained popularity in both the synchronous and asynchronous setting [30, 31]. However, those approaches assume decentralized workers are either completely synchronous (as in Fig. 1 (b)) or completely asynchronous, which requires to either deal with long execution time every iteration or pay for uncontrolled gradient staleness.

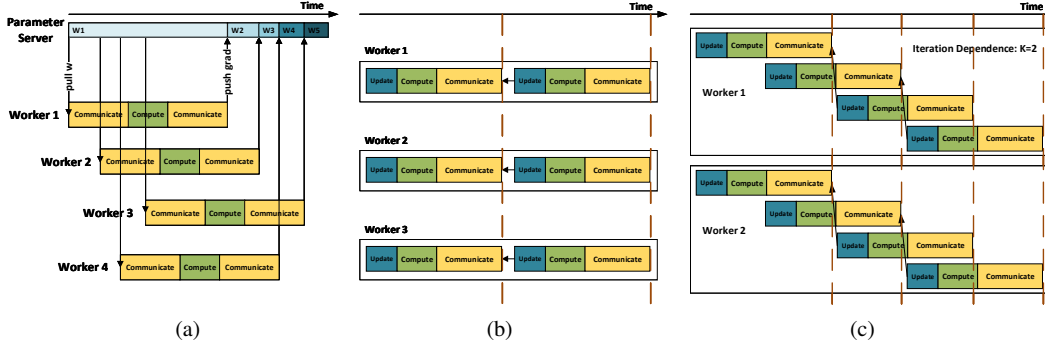

Figure 1: Comparison between different distributed learning frameworks: (a) parameter server with asynchronous training, (b) decentralized synchronous training, and (c) decentralized pipeline training.

**Compression in Distributed Training:** As the model size increases and cluster size scales, communication overhead in distributed learning system dominates the training time, *e.g.*, up to $80 \sim 90\%$ even in a high-speed network environment [29, 10]. To reduce the communication time, various compression algorithms have been proposed recently [43, 46, 11, 4, 50, 33, 2], some of which focus on reducing the precision of communicated gradients through scalar quantization into 1 bit, while others focus on reducing the quantity of gradients to be transferred. Most compression works, however, only emphasize on achieving high compression ratio or low loss in accuracy without reporting the wall-clock training time.

In practice, compression without knowledge of the communication process is usually counterproductive [29], *i.e.*, the total training time often increases. This is due to the fact that AllReduce is a multi-step algorithm which requires transferred gradients to be compressed and decompressed repeatedly with a worst-case complexity linear in the cluster size, as we discuss below in Sec. 3.2.

## 3 Decentralized Pipelined Stochastic Gradient Descent

**Overview:** To address the aforementioned issues (network congestion for a central server, long execution time for synchronous training, and stale gradients in asynchronous training) we propose a new decentralized learning framework, Pipe-SGD, shown in Fig. 1 (c). It balances communication among nodes via AllReduce and pipelines the local training iterations to hide communication time.

We developed Pipe-SGD by analyzing a timing model for wall-clock train time under different resource conditions using various communication approaches. We find that the proposed Pipe-SGD is optimal when gradient updates are delayed by only one iteration and the time taken by each iteration is dominated by local computation on workers. Moreover, we found lossy compression to further reduce communication time without impacting accuracy.

Due to local pipelined training, balanced communication, and compression, the communication time is no longer part of the critical path, *i.e.*, it is completely masked by computation, leading to linear speedup of end-to-end training time as the cluster size scales. Finally, we prove the convergence of Pipe-SGD for convex and strongly convex objectives by adjusting the proof of [23, 15].

### 3.1 Timing Models and Decentralized Pipe-SGD

**Timing Model:** We propose timing models based on decentralized synchronous SGD to analyze the wall-clock runtime of training. Each training iteration consists of three major stages: model update, gradient computation, and gradient communication. Classical synchronous SGD (Fig. 1 (b)) runs local iterations on workers sequentially, *i.e.*, each update depends on the gradient from the previous iteration, *i.e.*, the iteration dependency is 1. Therefore the total runtime of synchronous SGD can be formulated easily as:

$$l_{\text{total\_sync}} = T \cdot (l_{\text{up}} + l_{\text{comp}} + l_{\text{comm}}), \tag{2}$$

where $T$ denotes the total number of training iterations and $l_{\text{up}}, l_{\text{comp}}, l_{\text{comm}}$ refer to the time taken by update, compute, and communication, respectively. It is apparent that synchronous SGD depends on the sum of execution time taken by all stages, which leads to long end-to-end training time.

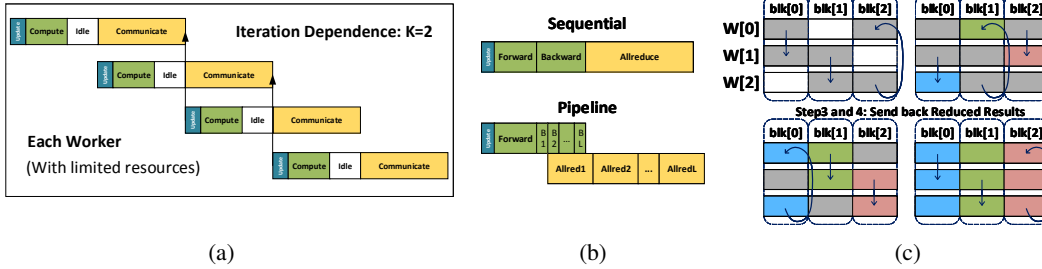

(a)                      (b)                     (c)

Figure 2: Timing model of Pipe-SGD: (a) each worker with limited resources, (b) sequential v.s. pipelined gradient communication, and (c) an example of gradient communication: Ring-AllReduce.

On the contrary, Pipe-SGD relaxes the iteration dependency to $K$, *i.e.*, each update depends only on the gradients of the $K$-th last iteration. This enables interleaving between neighboring iterations while maintaining globally synchronized communication, as shown in Fig. 1 (c). If we assume ideal conditions where both computation resources (CPU, GPU, other accelerators) and communication resources (communication links) are unlimited or abundant in counts/bandwidth, then the total runtime of Pipe-SGD is:

$$l_{\text{total\_pipe}} = T/K \cdot (l_{\text{up}} + l_{\text{comp}} + l_{\text{comm}}), \tag{3}$$

where $K$ denotes the iteration dependency or the gradient staleness. We observe that the end-to-end training time in Pipe-SGD can be shortened by a factor of $K$. However, the ideal resource assumption doesn't hold in practice, because both computation and communication resources are strictly limited on each worker node in today's distributed systems. As a result, the timing model for distributed learning is resource bound, either communication or computation bound, as shown in Fig. 2 (a), *i.e.*, the total runtime is:

$$l_{\text{total\_pipe}} = T \cdot \max(l_{\text{up}} + l_{\text{comp}}, l_{\text{comm}}), \tag{4}$$

where the total runtime is solely determined by either computation or communication resources, regardless of $K$ (when $K \geq 2$). Also, since gradient updates are always delayed by $(K-1)$ iterations, increasing $K > 2$ only harms, *i.e.*, the optimal value of $K = 2$ for Pipe-SGD with limited resources. Hence, the staleness of gradients is limited to 1 iteration, *i.e.*, the minimal staleness achievable in asynchronous updates. Besides, we generally prefer a computation-bound setting for distributed training system, *i.e.*, $l_{\text{up}} + l_{\text{comp}} > l_{\text{comm}}$. To achieve this we discuss compression techniques in Sec. 3.2.

In addition to pipelined execution of iterations, we also analyze pipelined gradient communication within each iteration to reduce train time. Computation of gradients, *i.e.*, the backward-pass, and communication of gradients are often executed in a strictly sequential manner (see Fig. 2 (b)). However, pipelined gradient communication, *i.e.*, communicating gradients immediately after they are computed, is feasible. Again, we assume limited resources and compare the sequential and pipelined gradient communication in Fig. 2 (b).

To analyze the detailed timing of the those two approaches, we use the timing models for communication [47]. Communication of gradients is an AllReduce operation which aggregates the gradient vector from all workers, performs the sum reduction element-wise, and then sends the result back to all. In practice, the underlying algorithms are much more involved [47]. For example, Ring-AllReduce, one of the fastest AllReduce algorithms, performs gradient aggregation collectively among workers through balanced communication. As shown in Fig. 2 (c), each worker transmits only a block of the entire gradient vector to its neighbor and performs the sum reduction on the received block. This "transmit-and-reduce" runs in parallel on all workers, until the gradient blocks are fully reduced on a worker (different for each block). Afterwards those fully reduced blocks are sent back to the remaining workers along the virtual ring. This approach optimally utilizes the network bandwidth of all nodes.

Adopting the Ring-AllReduce model of [47], we obtain the total runtime of Pipe-SGD with sequential gradient communication under the limited resource assumption via:

$$l_{\text{total\_pipe\_s}} = T \cdot \max\left(l_{\text{up}} + l_{\text{for}} + l_{\text{back}}, \ 2(p-1) \cdot \alpha + 2(\frac{p-1}{p}) \cdot n \cdot \beta + (\frac{p-1}{p}) \cdot n \cdot \gamma + S\right), \tag{5}$$

**Algorithm 1:** Decentralized Pipe-SGD training algorithm for each worker.

---
On the computation thread of each worker:

---
1: Initialize by the same model $w[0]$, learning rate $\gamma$, iteration dependency $K$, and number of iterations $T$.
2: **for** $t = 1, \ldots, T$ **do**
3:      Wait until aggregated gradient $g_{\text{sum}}^{\text{c}}$ in compressed format at iteration $[t - K]$ is ready
4:      Decompress gradient $g_{\text{sum}}[t - K] \leftarrow \text{Decompress}(g_{\text{sum}}^{\text{c}}[t - K])$
5:      Update $w[t] \leftarrow w[t - 1] - \gamma \cdot g_{\text{sum}}[t - K]$
6:      Load a batch $\mathscr{B}$ of training data
7:      Forward pass to compute current loss $f_{\mathscr{B}}$
8:      Backward pass to compute gradient $g_{\text{local}}[t] \leftarrow \frac{\partial f_{\mathscr{B}}}{\partial w[t]}$
9:      Compress gradient $g_{\text{local}}^{\text{c}}[t] \leftarrow \text{Compress}(g_{\text{local}}[t])$
10:     Denote local gradient $g_{\text{local}}^{\text{c}}[t]$ as ready
11: **end for**

---
On the communication thread of each worker:

---
1: Initialize aggregated gradients $g_{\text{sum}}^{\text{c}}$ of iteration $[1 - K, 1 - K + 1, \ldots, 0]$ as zero and mark them as ready
2: **for** $t = 1, \ldots, T$ **do**
3:      Wait until local gradient $g_{\text{local}}^{\text{c}}[t]$ is ready
4:      AllReduce $g_{\text{sum}}^{\text{c}}[t] \leftarrow \sum g_{\text{local}}^{\text{c}}[t]$
5:      Denote aggregated gradient $g_{\text{sum}}^{\text{c}}[t]$ as ready
6: **end for**

---

where $l_{\text{for}}$ and $l_{\text{back}}$ denote forward-pass and backward-pass time, $p$ denotes the number of workers, $\alpha$ the network latency, $n$ the model size in bytes, $\beta$ the byte transfer time, $\gamma$ the byte sum reduction time, and $S$ the global synchronization time.

Similarly, we obtain the total runtime of Pipe-SGD with pipelined gradient communication via:

$$l_{\text{total\_pipe\_p}} = T \cdot \max \left( l_{\text{up}} + l_{\text{for}} + l_{\text{b}}, \ 2(p-1)L \cdot \alpha + 2(\frac{p-1}{p}) \cdot n \cdot \beta + (\frac{p-1}{p}) \cdot n \cdot \gamma + L \cdot S \right), \quad (6)$$

where $L$ denotes the number of gradient segments, and $l_{\text{b}}$ denotes the backward-pass time taken by the first segment.

Based on Eq. (5) and Eq. (6) we note: if a pipelined system remains communication bound, then sequential gradient communication is preferred over the pipelined gradient communication (Eq. (5) is smaller than Eq. (6) due to positive $L$). In practice, distributed training of large models is often communication bound, making sequential exchange the best option.

To sum up, based on our timing models, we find: **Pipe-SGD is optimal for $K = 2$, system is compute bound (after compression), and sequential gradient communication is used**. Note that although our model is derived based on the Ring-AllReduce, this conclusion also applies to other AllReduce algorithms, such as recursive doubling, recursive halving and doubling, pairwise exchange, *etc.* [47].

**Decentralized Pipeline SGD:** Guided by the timing models, we develop the decentralized Pipe-SGD framework illustrated in Fig. 1 (c) where neighboring training iterations on workers are interleaved with a width of $K = 2$ while the execution within each iteration remains strictly sequential. Decentralized workers perform pipelined training in parallel with synchronization on gradient communication after every iteration. Due to the synchronous nature of our framework, the gradient update is always delayed by $K - 1$ iterations, which enforces a deterministic rather than an uncontrolled staleness. In our optimal setting, the number of iterations for a delayed update is 1, as compared to $O(p)$ where $p$ is the cluster size in the conventional asynchronous parameter server training [15, 31, 1]. Importantly, our framework still enjoys the advantage of an asynchronous approach – interleaving of training iterations to reduce end-to-end runtime. Also, different from the parameter server architecture, we don't congest the head node. Instead, in our case, every worker is only responsible for aggregating part of the gradients in a balanced manner such that communication and aggregate operation time are much more scalable.

More formally, we outline the algorithmic structure of our implementation for each worker in Alg. 1. To be specific, each worker has two threads: one for computation and one for communication, where the former thread consumes the aggregated gradient of the $K$-th last iteration and generates the local gradient to be communicated, and the latter thread exchanges the local gradient and buffers the aggregated results to be consumed by the former thread.

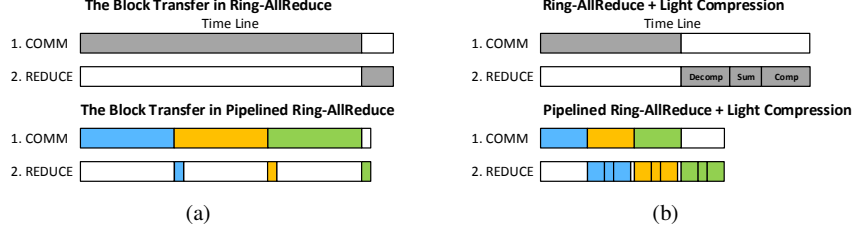

Figure 3: Pipelining within AllReduce: (a) block transfer in native Ring-AllReduce and pipelined Ring-AllReduce, and (b) block transfer with light-weight compression.

## 3.2 Compression in Pipe-SGD

To further reduce the communication time we integrate lossy compression into our decentralized Pipe-SGD framework. Unlike the conventional parameter server or recent decentralized framework transferring parameters over the network [8, 6, 27, 28, 36, 16, 15, 31, 30], our approach communicates only gradients and we justified empirically that gradients are much more tolerant to lossy compression than the model parameters. This seems intuitive since reducing the precision of parameters in every iteration harms the final precision of the trained model directly.

Importantly, as mentioned in Sec. 3.1, compressing the communication overhead contributes to the optimal setting of Pipe-SGD. Once Pipe-SGD is completely computation bound, linear speedups of end-to-end training time can be realized as the cluster size increases. Analytically, we show this observation by deriving the scaling efficiency using the timing model given in Eq. (4). Assume that: 1) the singe-node training takes $T_{single}$ iterations to complete with an execution time of $l_{single}$ taken by each iteration; 2) given a Pipe-SGD cluster with $p$ workers we use the same batch size on each worker as the single-node [12]; 3) the single node and Pipe-SGD train the same epochs on the dataset. From 2) and 3), we find that the total number of iterations required for Pipe-SGD is $T_{single}/p$, because Pipe-SGD has a $p$ times larger batch size while still training the same number of samples. From this we obtain the scaling efficiency *SE* of Pipe-SGD via

$$\text{SE} = \frac{\text{Actual Speedup}}{\text{Ideal Speedup}} = \frac{\frac{l_{single} \cdot T_{single}}{l_{total\_pipe}}}{p} = \frac{\frac{l_{single} \cdot T_{single}}{\max(l_{up} + l_{comp},\, l_{comm}) \cdot \frac{T_{single}}{p}}}{p} = \frac{l_{up} + l_{comp}}{\max(l_{up} + l_{comp},\, l_{comm})}. \quad (7)$$

Thus, we showed that once our system becomes compute bound with compressed communication, Pipe-SGD can achieve linear speedup as the cluster scales, *i.e.*, $SE = 1$.

To maintain applicability of Ring-AllReduce, we choose two simple compression approaches: truncation and scalar quantization. Truncation drops the less significant mantissa bits of floating-point values for each gradient. The scalar quantization discretizes each gradient value into an integer of limited bits, with a quantization range determined by the maximal element of a gradient vector. Due to their simplicity, we easily parallelize those compression approaches to minimize overhead.

Note that compression itself can be compute-heavy and the introduced computation overhead can outweigh the benefit of compressed communication. Particularly when considering that AllReduce based communication performs multiple steps to transfer and reduce the data (see Fig. 2 (c)), requiring repeated invocation of compression and decompression, *i.e.*, for each "transmit-and-reduce" step, with an invocation complexity linear in cluster size. Therefore, many proposed complex compression techniques [43, 46, 11, 4, 50, 33] often fail in the communication-optimal AllReduce setting, resulting in longer wallclock time. For these reasons, compression embedded inside AllReduce must be light, fast and easy to parallelize, such as a floating-point truncation or our element-wise quantization.

Indeed, *pipelining within AllReduce* can help alleviate the heavy overhead of complex compression. However, its benefit might still be limited. Instead of pipelining of training iterations as in Pipe-SGD, *pipelining within AllReduce* interleaves the gradient communication and reduction within each AllReduce process, as illustrated in Fig. 3 (a). Since the communication time is often larger than the reduction time, the latter can be hidden by the former. Once compression is used (as in Fig. 3 (b)), the two stage pipeline becomes (decompression, sum, compression) and (compressed communication) such that light compression overhead can be masked completely. Although complex compression may also benefit from the pipelined AllReduce, the improvement is limited because the time spent by complex compression often outweighs the communication time. For example,

we implemented [50] within the pipelined AllReduce and found that the compression overhead is $1.6 \sim 2.3\times$ the uncompressed communication time and $25.6 \sim 36.8\times$ the compressed communication time for the benchmarks in Sec. 4, in which case the heavy overhead cannot be masked. Complete masking requires the compression overhead to be smaller than the compressed communication. In the remainder, we only consider light compressions (truncation/quantization) with native AllReduce.

### 3.3 Convergence

To prove the convergence of Pipe-SGD we adapt the derivation from parameter-server based asynchronous training [15, 23]. We can show that the convergence rate of Pipe-SGD for convex objectives via SGD is $8FL\sqrt{\frac{K}{T}}$, where $K = 2$, $F$ and $L$ are constants for gradient distance and Lipschitz continuity, respectively. We can also show the convergence of Pipe-SGD for strongly convex functions, and find a rate of $O(\frac{\log T}{T})$ for gradient descent. These rates are consistent with [15, 23]. Due to the page limit we defer details to the supplementary material.

## 4  Experimental Evaluation

In this section, we demonstrate the efficacy of our approach on four benchmarks using three datasets: MNIST [26], CIFAR100 [21] and ImageNet [42]. We briefly review characteristics of those datasets before discussing metrics and setup, and finally presenting experimental results and analysis.

### Datasets and Deep Net Architecture

- `MNIST`: The MNIST dataset consists of 60,000 training and 10,000 test images, each showing one of ten possible digits. The images are of size $28 \times 28$ pixels with digits located at the center of the images. We use a classical 3-layer perceptron, MNIST-MLP, with both hidden layers being 500-dimensional and with a global batch size of 100.

- `ImageNet`: For our experiments we use 1,281,167 training and 50,000 validation examples from the ImageNet challenge. Each example comprises a color image of $256 \times 256$ pixels and belongs to one of 1000 classes. We use the classical AlexNet [22] and ResNet [14], both with a global batch size of 256.

- `CIFAR100`: The CIFAR100 dataset is composed of 50,000 training and 10,000 test examples with 100 classes. The simple AlexNet-style CIFAR100 architecture in [32] is used for benchmarking this datasets. It consists of 3 convolutional layers and 2 fully connected layers followed by a softmax layer. The detailed parameters are available in [32]. Importantly, we adapt this 5 layer CIFAR100-CNN into a convex optimization benchmark, CIFAR100-Convex, to match our proof of convergence. The convexity is achieved by training only the last fully connected layer while fixing the parameters of all previous layers.

### Metrics and Setup

We measure the wall-clock time of end-to-end training, *i.e.*, the same number of iterations for different settings. For each benchmark, we evaluate the timing model we proposed using end-to-end train time and detailed timing breakdowns. We plot the test/validation accuracy over training time to evaluate the actual convergence. Also, final top-1 accuracies on the test/validation set are reported. For the setup, we use a cluster of four nodes, each of which consists of a Titan XP GPU [40] and a Xeon CPU E5-2640 [17]. We employ an additional node as the parameter server to support the conventional centralized design. All nodes are connected by 10Gb Ethernet. We implement a distributed training framework in C++ using CUDA 8.0 [39], MKL 2018 [18], and OpenMPI 2.0 [41], which supports the parameter-server and Pipe-SGD approach.

### Results and Analysis

We evaluate the performance of three different frameworks: parameter server with synchronous SGD (PS-Sync), decentralized synchronous SGD (D-Sync), and Pipe-SGD. Our compression schemes, *i.e.*, 16-bit truncation (T) and 8-bit quantization (Q), are also applied to AllReduce communication in D-Sync and Pipe-SGD. Evaluation results are summarized in Fig. 4 where the first two columns show the convergence performances and the third column shows detailed timing breakdowns with final accuracies labeled.

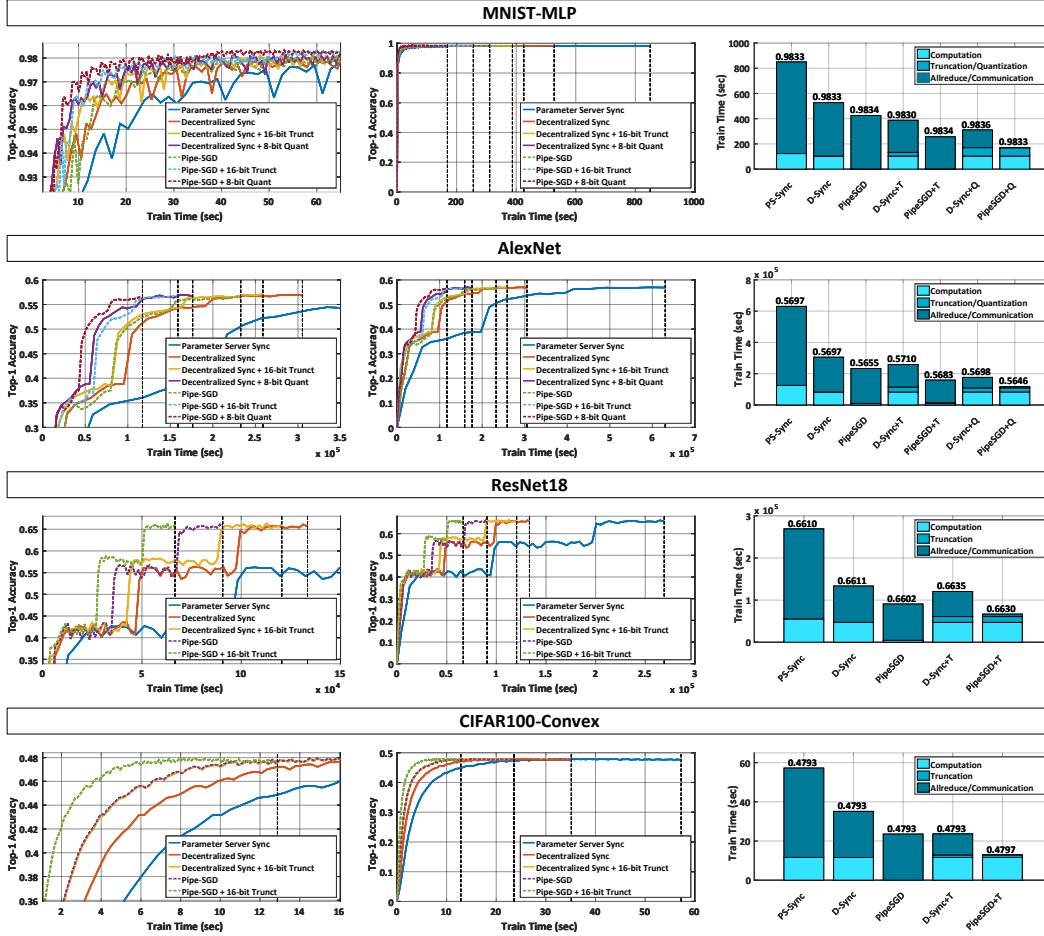

Figure 4: **Experimental results:** Each row shows different benchmarks. The left two columns show convergence via test/validation accuracy *vs.* wallclock training time, where the first column is an inset of the second one. The right most column shows the detailed timing breakdown of end-to-end training. Note that the final top-1 accuracies on test/validation set are labeled on top of the bars.

*Convergence*: From Fig. 4, we observe: decentralized approaches, *i.e.*, D-Sync and Pipe-SGD, converge much faster than the parameter server even without compression, and Pipe-SGD shows the fastest convergence among these frameworks, especially when compression is applied. For example, the convergence curve of the CIFAR100-Convex shows that D-Sync is around 40% faster than PS-Sync and Pipe-SGD is another 37% faster than D-Sync. The advantage of Pipe-SGD is further boosted by compression, *i.e.*, truncation in this case, and demonstrates an additional 46% faster convergence than the D-Sync with the same compression scheme. Therefore Pipe-SGD prevails with a great margin.

*Timing Breakdown*: From Fig. 4, the comparison between centralized and decentralized designs shows 50% reduction in uncompressed communication time, thus justifying the efficacy of balanced communication. Once compression is applied, further reduction is observed. However, the actual improvement in D-Sync is not ideal considering compression factors of 2× for truncation and 4× for quantization, because the compression overhead is paid at the critical path of D-Sync. In contrast, our Pipe-SGD can hide this overhead together with computation due to the pipelined nature, as shown in "D-Sync+T" *vs.* "PipeSGD +T" in the MNIST benchmark. As communication is further reduced by quantization, the system becomes compute bound and Pipe-SGD switches to hide the communication instead, thus reaching the optimal setting of Pipe-SGD. This optimum can also be achieved via the simplest truncation for models with less dominant communication time, *e.g.*, ResNet18 and CIFAR100-Convex. As a result, our approach achieves a speedup of 2.0 ∼ 3.2× compared to D-Sync and 4.0 ∼ 5.4× compared to PS-Sync for these benchmarks. Note that these speedups are based on the comparison between different approaches in the same cluster without scaling the cluster size.

*Accuracy*: Considering the potential drawback of the 1-iteration staled update and lossy compression in Pipe-SGD, we also evaluate the final test/validation accuracies after end-to-end training, as shown in Fig. 4. Interestingly, in our optimal settings "PipeSGD +T/Q," we find that only AlexNet drops top-1 accuracy by 0.005 compared to baseline D-Sync while all other benchmarks show slightly improved accuracies. To obtain the best accuracies for the two large non-convex models such as AlexNet and ResNet, we employ a similiar warm-up scheme as in [33], *i.e.*, we don't turn on the pipelined training until the 5-th epoch, before which we still stick to D-Sync training to avoid the undesirable gradient change in the initial stage. Since the warm-up period is marginal compared to total number of epochs, the system performance benefits from Pipe-SGD most of the time. Note that for smaller models, especially convex ones (*e.g.*, CIFAR100-Convex), no warm-up is required.

## 5    Related Work

Li *et al*. [27, 28] proposed a parameter server framework for distributed learning and a few approaches to reduce the cost of communication among compute nodes, such as exchanging only nonzero parameter values, local caching of index list, and random skip of messages to be transmitted. Abadi *et al*. [1] also proposed a centralized framework, TensorFlow, which incorporates model and data parallelism for training deep nets. Both works support the asynchronous setting to improve communication efficiency but without controlling the staleness of the gradient update. Ho *et al*. [15] proposed SSP, another centralized asynchronous framework but with bounded staleness for gradients. The key idea of SSP: 1) each worker has its own iteration index, 2) the slowest and fastest worker must be within $S$ iterations, otherwise, the fastest worker is forced to wait until the slowest worker catches up. However, this bound $S$ applies to the iteration drift among workers instead of directly on the stale updates of the parameter server. As a result, each worker within the bound can still commit their updates to the server asynchronously, making the last gradient update staled heavily. In the worst case, the staleness is linear in the cluster size.

Lin *et al*. [33] employed AllReduce as the gradient aggregation method in their synchronous framework, but little is reported regarding wallclock time benefits, especially considering that the full synchronous design suffers from the longest execution time among all workers. Besides, Lian *et al*. proposed AD-PSGD [31] which parallelizes the SGD process over decentralized workers in a completely asynchronous fashion. Workers run completely independently, and only communicate with a set of neighboring nodes to exchange trained weights, *i.e.*, neighboring models are averaged to replace each worker's local model in each iteration. However, this approach suffers from uncontrolled staleness, which in practice increases with cluster size and the time taken by each iteration. In addition, such a communication method requires each worker to act as the center node of a local graph, which results in a local communication bottleneck. As a result, each worker suffers from long iteration time which further increases the staleness of weight updates. Although Lian *et al*. [31] compared their framework with the full synchronous design in wall-clock time, the performance turns out to be similar when network speeds are roughly equal.

Recently, independent work [13] also proposed a distributed pipelined system for DNN training. Different from Pipe-SGD, [13] focuses on pipelining with model parallelism, partitioning the DNN layers onto different machines and pipelining the execution of the machines by injecting consecutive mini-batches into the first one. This approach reduces communication load since only activations and gradients of a subset of layers are communicated between machines. However, complex mechanisms (such as profiling, partitioning algorithm, and replicated stages) are necessary to balance the workload among different machines, otherwise compute resources turn idle. Furthermore, [13] may suffer from staleness of the weight update, which is linear in the number of stages. This limits the effectiveness of model pipelining and throttles speedups.

## 6    Conclusion

We developed a rigorous timing model for distributed deep net training which takes into account network latency, model size, byte transfer time, *etc*. Based on our timing model and realistic resource assumptions, *e.g*., limited network bandwidth, we assessed scalability and developed Pipe-SGD, a pipelined training framework which is able to mask the faster of computation or communication time. We showed efficacy of the proposed method on a four-node GPU cluster connected with 10Gb links. Rigorously assessing wall-clock time for Pipe-SGD, we are able to achieve improvements of up to $5.4\times$ compared to conventional approaches.

## Acknowledgement

This work is supported in part by grants from NSF (IIS 17-18221, CNS 17-05047, CNS 15-57244, CCF-1763673 and CCF-1703575). This work is also supported by 3M and the IBM-ILLINOIS Center for Cognitive Computing Systems Research (C3SR). Besides, this material is based in part upon work supported by Defense Advanced Research Projects Agency (DARPA) under Contract No. HR001117C0053. The views, opinions, and/or findings expressed are those of the author(s) and should not be interpreted as representing the official views or policies of the Department of Defense or the U.S. Government.

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
