[Supplementary Material]

**Supplementary Material — Pipe-SGD: A Decentralized Pipelined SGD Framework for Distributed Deep Net Training**

## A   Proof of Convergence

To prove the convergence of Pipe-SGD for convex objective functions we adapt the derivation from parameter-server based asynchronous training [23, 15].

Specifically, by inserting $K-1$ virtual states of the parameters before the first actual parameter update in Alg. 1, we can mathematically formulate the parameter update in Pipe-SGD as follows:

**Proposition 1** *In Pipe-SGD with iteration dependency of K, parameters are updated as:*

$$\begin{cases} w[1] \triangleq \text{initial parameters} \\ w[t] = w[1], & t \in [2, K] \\ w[t] = w[t-1] - \eta[t-1]g[t-K], & t \in [K+1, T], \end{cases} \tag{8}$$

*In other words, there is a constant delay of $K-1$ steps between gradient computation and its application for a weight update.*

We note that this update is identical to that of the parameter-server proposed in [23], in which $K$ workers upload their gradients to the server in a round-robin manner, and the parameters are updated using the gradient uploaded $K-1$ iterations ago.

Therefore, the convergence of Pipe-SGD can be derived using the same methods applied in [23, 15]. For instance, using [23], we confirm the convergence of Pipe-SGD for strongly convex functions, and find a rate of $O(\frac{\log T}{T})$ for gradient descent. Moreover, we can reduce the above parameter update model to a special case in SSP [15] with $K$ workers and an iteration drift factor of $s = 1$ to show that the convergence rate of Pipe-SGD training for standard convex functions via SGD is $8FL\sqrt{\frac{K}{T}}$, where $F$ and $L$ are constants regarding gradient distance and Lipschtiz continuous, respectively.