[Reviews · NeurIPS 2018]

Reviewer 1



This paper proposed a pipelined training setup for neural nets. The pipeline structure allows computation and communication to be carried out concurrently within each worker. Experiments show speed improvements over existing methods. Overall I think the method and the evaluations are convincing. However, I believe the NIPS conference is not the appropriate venue for this paper for the following reasons. This paper deals with a high performance computing problem, namely, how to get more throughput (computation) in a distributed network. This is a low level computer systems problem that does not involve machine learning, statistics or neural science. As much as it is designed for deep neural nets, the nature of the approach is not really about improving SGD itself or neural nets architecture or coming up with a new loss. In summary, I do believe this work has the potential of changing the basic library for distributed neural nets training because of its improved performance. I just feel that this paper would fit better to other high performance computing conferences, like many of the cited related work.

Reviewer 2



This paper presents a collection of techniques which make parallel distributed SGD scale linearly - and exceed any preceding approaches. While the paper highlights the pipelined approach (overlapping computation of one worker with communication of another), the authors also present a decentralized algorithm that takes advantage of the pipeline approach. The work is well motivated by wall-clock timing, which the authors point out is important as some factors, like the cost of computation for compression, are often left out in parallel and decentralized optimization studies. The only thing really lacking in this work is a more detailed analysis of where the allreduce algorithm is spending it;s time. It's clear that the hierarchical sum is more efficient than a linear sum, but does this need to be done on separate machines (which invokes the need to compress and decompress at each level) - or could this be offset via other overlapping pipelined computation.

Reviewer 3



The paper develops a timing model for distributed DNN training based on AllReduce and, based on this, proposes a pipelined training strategy that allows to mask the faster of computation and communication times. The paper proves convergence in the convex and strongly convex case and evaluates PipeSGD empirically on a variety of benchmarks (MNIST, CIFAR, ImageNet) using a GPU cluster with 4 nodes. Empirically, PipeSGD leads to a significant speed-up relative to decentralized synchronous SGD and synchronous SGD. The paper is clearly written and as far as I can tell the timing model makes sense. The empirical results are quite favorable but I would encourage the authors to attempt to scale this framework to larger clusters: 4 GPUs is quite small for distributed deep learning and the paper is likely to have a bigger impact if PipeSGD can be shown to work well for larger cluster sizes.